# Contextual Bandits with LLM-Derived Priors and Adaptive Calibration

## Abstract

Large Language Models (LLMs) offer rich prior knowledge that can accelerate online decision-making, yet their use in contextual bandits lacks principled mechanisms for guiding exploration. We address this gap by proposing a lightweight framework that integrates LLM-derived priors with adaptive calibration in a multi-armed bandit setting. Our method first extracts prompt-based rewards from the LLM to provide task-specific supervision. We then construct an embedding-based estimator that quantifies uncertainty from the LLM's representations, yielding a calibrated exploration signal. To remain robust under distribution shifts, we introduce an online contextual adapter that dynamically updates these uncertainty estimates during interaction. Experiments on LastFM and MovieLens-1M show that our method consistently outperforms both classical bandits and pure LLM-based agents, achieving higher cumulative rewards with significantly fewer LLM queries. Furthermore, we provide theoretical regret guarantees that establish improved sample efficiency compared to standard contextual bandits.

## 1 Introduction

In online decision-making tasks such as adaptive tutoring, information routing, and interactive recommendation, systems must continually balance exploration and exploitation to maximize long-term outcomes (Huang et al., 2024a; Chen et al., 2024b; Borchers & Shou, 2025). Traditional approaches to these problems are often framed as contextual bandits or reinforcement learning, where effective exploration strategies are crucial for data-efficient learning (Chen et al., 2024b). Large language models (LLMs) introduce new opportunities in this setting: beyond serving as inference engines, they may accelerate learning by reasoning about context, generalizing from limited feedback, and adapting decisions in real time.

However, for these applications to be truly reliable, LLMs need robust mechanisms to estimate uncertainty (Abbasi Yadkori et al., 2024; Ma et al., 2025). This is essential for balancing exploration and exploitation, ensuring the model's robustness under distribution shifts, and supporting data-efficient learning (Zhang et al., 2023; Deng & Raffel, 2023). Despite their remarkable performance, current approaches to uncertainty estimation in LLMs often rely on heuristic methods such as temperature scaling, response entropy, or sampling-based diversity measures (Huang et al., 2024b). While these techniques can be useful in specific scenarios, they lack calibration, fail to generalize across different tasks, and offer limited insights into the model's confidence levels. Other more sophisticated methods involve fine-tuning or ensemble approaches, which are computationally intensive and not suitable for general-purpose, pre-trained LLMs (Krishnan et al., 2024).

To address these limitations, we propose a lightweight and flexible framework for uncertainty-aware decision-making with LLMs. This framework includes two key components: (1) a prompt reward, which extracts supervision from the model's outputs by leveraging internal knowledge, and (2) a last-layer embedding-based uncertainty estimator that captures semantic confidence. Additionally, our approach adapts to distribution shifts using an online contextual uncertainty adapter, allowing for dynamic calibration in changing environments. We demonstrate this framework within a multi-armed bandit (MAB) setting, integrating it with exploration strategies through calibrated Upper Confidence Bound/Lower Confidence Bound estimates.

The LLM and bandit integration is non-trivial due to the inherent challenges of mapping high-dimensional, task-agnostic LLM embeddings into actionable uncertainty estimates. These challenges

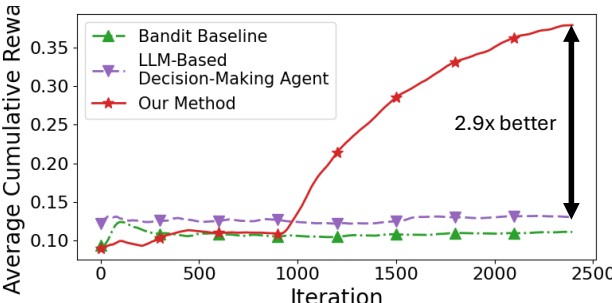

Figure 1: Performance comparison on the LastFM recommendation benchmark, comparing (i) a conventional bandit baseline, (ii) an LLM-based decision agent, and (iii) our calibrated LLM-MAB method. Further implementation details are provided in Section 5.1.

include resolving semantic misalignment, addressing representation drift, and ensuring consistent scale. Moreover, integrating these uncertainty estimates into structured decision-making frameworks, such as MABs, requires careful calibration to maintain theoretical guarantees while preserving the efficiency of pre-trained LLMs. Our method successfully addresses these complexities while remaining lightweight and compatible with zero-shot applications. To summarize, our key contributions are:

- We propose a lightweight LLM-calibrated agent that combines prompt-derived supervision with an embedding-based uncertainty estimator to guide multi-arm bandit exploration.

- We demonstrate an LLM-enabled enhancement to exploration strategies: dynamically adjusting conservative UCB estimates during early learning phases.

- We validate our framework on multiple benchmarks, achieving state-of-the-art cumulative reward while significantly reducing expensive LLM queries compared to pure LLM and classical bandit baselines (See Figure 1).

- We provide theoretical guarantees for the proposed method that quantifies the regret bound on online learning, which demonstrates improvement over pure LLM-based prediction or classic algorithms such as LinUCB.

**Paper Outline:** The remainder of this paper is organized as follows. Section 2 reviews related work on contextual bandits and bandits in LLMs. Section 3 introduces the preliminaries and formalizes our problem setup. Section 4 presents our methodology, including prompt reward extraction, embedding-based uncertainty quantification, and integration with multi-armed bandits, along with theoretical analysis. Section 5 describes the experimental setup and results, covering datasets, baselines, implementation details, ablation studies, and computational efficiency. Section 6 concludes with a summary of contributions, while the appendix provides additional case studies and theoretical proofs.

## 2 RELATED WORK

### 2.1 CONTEXTUAL BANDITS

Contextual bandits (CB) formalize sequential decision-making with side information under bandit feedback, balancing exploration and exploitation to maximize cumulative reward. Foundational reductions and algorithms established the modern landscape. Agarwal et al. (2014b) reduced CB to cost-sensitive classification with provable guarantees and practical policies; Langford & Zhang (2008) analyzed Epoch-Greedy as a simple exploration strategy. Linear models are central: Li et al. (2010a) introduced LinUCB with large-scale deployment on Yahoo news recommendation, while Abbasi-Yadkori et al. (2011a) provided tighter confidence-based analysis OFUL for stochastic linear rewards. Beyond linearity, Filippi et al. (2010) studied generalized linear bandits, and kernel methods such as GP- and RKHS-based UCB extended CBs to rich function classes (Valko et al., 2013). Bayesian and Thompson sampling perspectives offer complementary solutions. Agrawal & Goyal (2013) established regret guarantees for Thompson Sampling in linear bandits, and subsequent work refined practical implementations for large-scale problems (Riquelme et al., 2018). As representation

learning became critical, neural variants emerged: NeuralUCB and NeuralTS control optimism or posterior uncertainty over deep features to achieve sublinear regret in non-linear regimes while remaining computationally feasible (Zhou et al., 2020). Practical CB systems must also handle constraints and dynamics. Agarwal et al. (2014a) introduced Contextual Bandits with Knapsacks to manage resource and fairness constraints, while non-stationary environments have been studied via variation-budget and change-point models to maintain low regret under drift (Besbes et al., 2014).

## 2.2 LLM-BASED BANDITS

The integration of large language models (LLMs) with online learning and bandit algorithms has recently attracted increasing attention. Early studies recognized the synergy between bandits and LLMs: bandit methods provide exploration–exploitation guarantees, while LLMs offer contextual reasoning and prior knowledge. The survey of Bouneffouf & Feraud (2025) systematically outlines this emerging connection, in terms of both how bandit algorithms can enhance the efficiency and adaptability of LLMs, and how LLMs can contribute to the contextual and adaptive decision-making capabilities of bandit algorithms. Monea et al. (2024) found that LLMs exhibit in-context reinforcement–learning behavior, providing a mechanism by which few-shot prompts can induce sequential improvement without gradient updates. Alamdari et al. (2024) showed empirically that LLM-generated prior knowledge can "jump-start" exploration by supplying informative priors, thereby accelerating learning in early rounds. Xia et al. (2024) studied the problem of in-context dueling bandits with LLM agents, demonstrating that preference-based feedback can be leveraged beyond numeric rewards to facilitate the combination. Beyond these methodological research, the combination of LLM with bandits also witnesses application-driven success, such as LLM-tailored health messaging (Song et al., 2025), online marketing (Ye et al., 2025), among others.

## 3 PRELIMINARIES

**Notations.** Let $n \in \mathbb{N}_+$ be a positive integer. $[n]$ denotes the set $\{1, ..., n\}$. For any set $\mathcal{S}$, $|\mathcal{S}|$ denotes the number of elements in $\mathcal{S}$. For vector norms, $\|\boldsymbol{x}\|_p$ denotes the $\ell_p$ norm of vector $\boldsymbol{x}$.

**Problem Formulation.** We formulate the problem as a contextual decision-making task, where the expected reward of an action depends on both user and item contexts. The goal of the learning agent is to maximize the cumulative expected reward over time. Let $\mathcal{I} := \{1, \ldots, I\}$ denote a finite set of items (e.g., movies, articles) to be recommended, referred to as arms, where $I \in \mathbb{N}_+$. Each item $i \in \mathcal{I}$ is associated with a context vector $\boldsymbol{c}_i \in \mathcal{C}$, where the context space $\mathcal{C}$ represents a subspace of the language space (e.g., text descriptions or prompts). The learning agent interacts with users over $T \in \mathbb{N}_+$ rounds and has access to a large language model, denoted as LLM. In each round $t = 1, 2, \ldots, T$, the agent observes the context $x_t \in \mathcal{C}$ of the incoming user and a candidate subset of arms $\mathcal{I}_t \subseteq \mathcal{I}$. Based on the user context $x_t$, item contexts $\{c_i\}_{i \in \mathcal{I}_t}$, and past interactions, the agent constructs a prompt $p_t$ to query the LLM and receives a textual response $o_t \sim \text{LLM}(p_t)$. Using $o_t$ and potentially other relevant information (e.g., offline user-item profiling), the agent selects an item $i_t \in \mathcal{I}_t$ to recommend to the user.

The agent then receives a reward $r_t \in [0, 1]$, with expected value $\mathbb{E}[r_t] = f(x_t, c_{i_t})$. The reward function $f$ is unknown and will be further specified below. Let $i_t^* = \arg\max_{i \in \mathcal{I}_t} f(x_t, c_i)$ denote the optimal arm at round $t$ that yields the highest expected reward. The agent's performance is measured by the cumulative regret:

$$R(T) = \sum_{t=1}^{T} f(x_t, c_{i_t^*}) - \sum_{t=1}^{T} f(x_t, c_{i_t}). \tag{1}$$

**Reward Model.** In this work, we assume a linear contextual reward model, where there exists an embedding function $e : \mathcal{C} \times \mathcal{C} \to \mathbb{R}^d$ and an unknown user preference vector $\boldsymbol{\theta}^* \in [0, 1]$, such that:

$$r_t = e(x_t, c_{i_t})^\top \boldsymbol{\theta}^* + \varepsilon_t, \tag{2}$$

where $\varepsilon_t$ is a zero-mean 1-sub-Gaussian noise term. The function $f$ is thus defined as $f(x_t, c_{i_t}) = e(x_t, c_{i_t})^\top \boldsymbol{\theta}^*$, and we assume $\|\boldsymbol{\theta}^*\|_2 \leq 1$. While our setting focuses on the linear model for clarity, our algorithm can be naturally extended to other settings such as logistic models (e.g.,

$f = \sigma(e(x_t, c_{i_t})^\top \boldsymbol{\theta}^*))$, kernelized reward models, or general non-linear approximators (e.g., neural networks). With a little abuse of notation, we may use $x_{t,i} = [x_t, c_i]$ to denote the joint context of user $t$ and item $i$.

# 4 METHODOLOGY

In this section, we describe our two-stage framework, comprising (i) an offline expert finetuning phase (Algorithm 1), and (ii) an online LLM–MAB integration phase (Algorithm 2). We detail our lightweight and flexible framework for uncertainty-aware decision-making with LLMs. Our approach comprises two complementary components: a *prompt reward* module that leverages internal LLM knowledge for supervisory signals, and a last-layer embedding-based uncertainty estimator that quantifies semantic confidence. We further introduce an online contextual uncertainty adapter, enabling dynamic calibration under distributional shifts. Finally, we describe how these elements integrate within a multi-armed bandit (MAB) setting, yielding calibrated exploration strategies.

## 4.1 PROMPT REWARD EXTRACTION

The first component of our framework harnesses the generative capabilities of pretrained LLMs to produce reward estimates without additional fine-tuning. Given a context $x \in \mathbb{R}^d$, we construct a natural language prompt that queries the LLM for an evaluation of the expected reward. The resulting probability or score is then normalized to fall in $[0, 1]$, forming the *prompt-derived reward* $r_{\text{prompt}}$. This mechanism exploits the LLM's internal representations and world knowledge to provide zero-shot supervision, circumventing the need for task-specific training while capturing rich semantic cues.

## 4.2 EMBEDDING-BASED UNCERTAINTY QUANTIFICATION

To accompany prompt rewards with calibrated uncertainty estimates, we introduce a lightweight estimator based on the penultimate-layer embeddings of the LLM. For each context $x$, we extract the last-layer feature vector $e(x) \in \mathbb{R}^{d_{\text{last}}}$ from the LLM. We then learn a linear projection (logistic regression) head $\phi : \mathbb{R}^{d_{\text{last}}} \to [0, 1]$ on offline dataset predicting the observed reward. Simultaneously, we accumulate the empirical covariance matrix:

$$\mathbf{V} = \sum_{i=1}^{n} e(x_i)e(x_i)^\top + \lambda \mathbf{I}, \tag{3}$$

where $\lambda$ is a regularization constant. At inference time, we compute a confidence interval around the point estimate via:

$$r_{\text{est}}(x) = \phi(e(x)), \quad u(x) = \beta\sqrt{e(x)^\top \mathbf{V}^{-1} e(x)}, \tag{4}$$

generating upper and lower confidence bounds $r_{\text{est}}(x) \pm \nu(x)$. This design captures semantic misalignment and representation drift by directly operating on high-dimensional embeddings, while remaining computationally efficient.

***Online Contextual Uncertainty Adapter.*** While the expert module is initially trained on an offline dataset, real-world deployments often encounter distributional shifts that deviate from the training distribution. Such shifts can degrade the calibration quality of the reward estimator and uncertainty quantifier. To address this, we introduce an online contextual uncertainty adapter. To be specific, we deploy an online adapter that periodically updates $\phi$ and $\mathbf{V}$ using newly observed $(e(x_t), r_t)$ pairs. This adapter leverages incremental updates to the covariance matrix and online least-squares adjustment of $\phi$, ensuring that uncertainty estimates remain accurate under distributional shifts. By blending offline pretraining with light online adaptation, our framework balances stability and flexibility.

## 4.3 INTEGRATION WITH MULTI-ARMED BANDITS

To effectively leverage both the semantic priors from prompt-based LLM supervision and the calibrated uncertainty estimates from embedding-based prediction, we integrate these signals into

a contextual multi-armed bandit (MAB) framework. This integration enables uncertainty-aware exploration that is both data-efficient in early rounds and adaptively grounded in observed feedback as more data becomes available. In particular, the LLM's prompt-derived reward offers informative, zero-shot guidance at the start of learning, while the embedding-based estimator provides increasingly accurate posterior estimates with calibrated uncertainty as the environment is explored. By fusing these two perspectives, we design a composite reward function that smoothly transitions from prior-driven to data-driven decision-making. We define the adjusted reward for each arm as:

$$\tilde{r}_i = \mathrm{clip}\big(r_{\mathrm{prompt}}(x_i); \, r_{\mathrm{est}}(x_i) - \nu(x_i),$$
$$r_{\mathrm{est}}(x_i) + \nu(x_i)\big). \tag{5}$$

During an initial exploration phase, actions are selected purely by maximizing $r_{\mathrm{prompt}}$. In the subsequent joint phase, the clipped reward $\tilde{r}_i$ drives selection, optionally augmented with classic UCB terms from a linear bandit on raw features. This calibrated fusion preserves theoretical exploration guarantees while capitalizing on pretrained LLM knowledge and uncertainty estimates. Experimental results in Section 5 demonstrate that this integration yields substantial improvements in cumulative reward and sample efficiency.

---

**Algorithm 1** Offline LLM Expert Finetuning

---

1: **Input:** Offline dataset $\mathcal{D} = \{(x_t, r_t)\}_{t=1}^n$, foundation model `LLM`, regularization factor $\lambda$.
2: **for** $t = 1, ..., n$ **do**
3:     Input $x_t$ into `LLM` and extract the second last layer's embedding $e_t(x_t, r_t) \in \mathbb{R}^{d_{\mathrm{last}}}$.
4: **end for**
5: Freeze all but the projection head $\phi$ (the last linear layer) of `LLM`, retrain $\phi$ through dataset $\mathcal{D}$.
6: **Return:** Projection head $\phi : \mathbb{R}^{d_{\mathrm{last}}} \to [0, 1]$, covariance matrix $\boldsymbol{V} = \sum_{t=1}^n e_t(x_t, r_t)e_t^\top(x_t, r_t) + \lambda \boldsymbol{I} \in \mathbb{R}^{d_{\mathrm{last}} \times d_{\mathrm{last}}}$.

---

**Algorithm 2** Online LLM and MAB integration (A1): calibrate LLM with MAB

---

1: **Input:** Finetuned `LLM`, projection layer $\phi$, covariance matrix $\boldsymbol{V}$, online decision rounds $T$, exploration phase $\sigma T$, coefficient $\beta_{\mathrm{LLM}}, \beta_{\mathrm{MAB}}$.
2: **Initialize:** MAB covariance matrix $\boldsymbol{G}_t = \lambda \boldsymbol{I} \in \mathbb{R}^{d \times d}$, and regressand $\boldsymbol{b}_t = 0 \in \mathbb{R}^d$.
3: **for** $t = 1, ..., T$ **do**
4:     User $u_t$ comes to the system with $K$ items with contexts $x_{t,1}, ..., x_{t,K} \in \mathbb{R}^d$
5:     Extract the second last layer's embedding $e_t(x_{t,i}) \in \mathbb{R}^{d_{\mathrm{last}}}$ for each item $i \in [K]$.
6:     Compute LLM reward prediction (1) estimated reward $r_{\mathrm{LLM}}(i) = $ Prompt, (2) UCB-style reward: $r_{\mathrm{LLM}}(i) = \phi^\top e_t(x_{t,i}) + \beta_{\mathrm{LLM}}\sqrt{e_t^\top(x_{t,i})\boldsymbol{V}^{-1}e_t(x_{t,i})}$, (3) LCB-style reward: $r_{\mathrm{LLM}}(i) = \phi^\top e_t(x_{t,i}) - \beta_{\mathrm{LLM}}\sqrt{e_t^\top(x_{t,i})\boldsymbol{V}^{-1}e_t(x_{t,i})}$.
7:     Compute UCB reward prediction for MAB: $r_{\mathrm{UCB}}(i) = \hat{\theta}_t^\top x_{t,i} + \beta_{\mathrm{MAB}}\sqrt{x_{t,i}^\top \boldsymbol{G}_t^{-1} x_{t,i}}$
8:     Compute LCB reward prediction for MAB: $r_{\mathrm{LCB}}(i) = \hat{\theta}_t^\top x_{t,i} - \beta_{\mathrm{MAB}}\sqrt{x_{t,i}^\top \boldsymbol{G}_t^{-1} x_{t,i}}$
9:     **if** $t \leq \sigma T$ **then**
10:         Select $i_t = \arg\max_{i \in [K]} r_{\mathrm{LLM}}(i)$.
11:     **else**
12:         Select $i_t = \arg\max_{i \in [K]} \tilde{r}(i)$, where $\tilde{r}(i) = \mathrm{CLIP}_{[r_{\mathrm{LCB}}(i), r_{\mathrm{UCB}}(i)]} r_{\mathrm{LLM}}(i)$.
13:     **end if**
14:     Receive reward $r_t$ for item $i_t$.
15:     Update MAB's statistics $G_{t+1} = G_t + x_{t,i_t}x_{t,i_t}^\top, b_{t+1} = b_t + x_{t,i_t}r_t$.
16: **end for**

---

### 4.4 THEORETICAL ANALYSIS

To formally characterize the performance of our LLM–MAB integration, we establish theoretical assumptions and supporting lemmas.

**Assumption 1** (Linear reward model). *Suppose the feature vectors $e_t(a) \in \mathbb{R}^d$ satisfy $\|\phi_t(a)\|_2 \leq L$. The reward $r_t \in [0,1]$ of the played arm has the following form*

$$r_t(a) = e_t(a)^\top \theta^\star + \varepsilon_t(a), \qquad \|\theta^\star\|_2 \leq S,$$

*and the noise $\varepsilon_t(a)$ are independent mean-zero subgaussian variables with parameter $R$.*

Denote the conditional mean of the reward $r_t(a)$ as $\mu_t(a)$.

**Assumption 2** (Random LLM scores: biased subgaussian). *For each round $t$ and arm $a \in \mathcal{A}_t$, the LLM score $s_t(a)$ is revealed before choosing $a_t$, and conditioned on the history $\mathcal{F}_{t-1}$ and the current context/candidate set,*

$$s_t(a) = \mu_t(a) + b_t(a) + \xi_t(a),$$

*where the* bias *satisfies $|b_t(a)| \leq b$ for some $b \geq 0$, and the* zero-mean *noise $\xi_t(a)$ is $\sigma_s$-subgaussian conditionally on $\mathcal{F}_{t-1}$ (i.e., $\mathbb{E}[e^{\lambda \xi_t(a)} \mid \mathcal{F}_{t-1}] \leq \exp(\frac{\lambda^2 \sigma_s^2}{2})$ for all $\lambda \in \mathbb{R}$). The sizes $A_t := |\mathcal{A}_t|$ are finite; define $A_{\max} := \max_{1 \leq t \leq T} A_t$.*

**Theorem 1** (Regret bound for Algorithm 2). *Fix $\alpha \in (0,1)$. Define*

$$\Delta_\alpha := b + \sigma_s \sqrt{2 \log\left(\frac{A_{\max} T}{\alpha}\right)}.$$

*Under Assumptions 1 and 2, define*

$$\Psi := d \log\left(1 + \frac{(T-T_0)L^2}{d \, \lambda_{\min}(V_{T_0})}\right), \quad B_T := R\sqrt{2\left(d \log(1 + \frac{(T-T_0)L^2}{d\lambda_{\min}(V_{T_0})}) + \log\frac{1}{\delta_0}\right)} + \sqrt{\lambda}\, S.$$

*For any $\delta_0 \in (0,1)$, with probability at least $1 - \delta_0 - \alpha$,*

$$R_T \leq \underbrace{2\Delta_\alpha T_0}_{\text{exploration}} + \underbrace{4 \min\left\{\Delta_\alpha(T-T_0), \, B_T\sqrt{2(T-T_0)\,\Psi}\right\}}_{\text{post exploration}}. \tag{6}$$

The proofs of the theorem are provided in Appendix C. Theorem 1 implies the following messages. First, the exploration phase induces a regret of $2\Delta_\alpha T_0$, which depends on the accuracy of the LLM measured by $\Delta_a$. It depends on two aspects: the inherent bias $b$ and the randomness of the LLM prediction $\sigma_s$. $\sigma_s$ can usually be reduced by using a lower temperature. $b$ depends on many factors, such as prompt, model, etc. If the LLM is predictive of the rewards, then $\Delta_\alpha$ is small, and the exploration phase is leading to smaller error. Second, the post-exploration phase induces a regret of the minimum of two terms. The first term is the potential benefits of using LLM-based predictions, similar to the exploration phase. The second part has the form of the classic linear bandit regret bound. Nevertheless, it is better than the classic linear bandit regret bound because the factor $B_T$ and $\Psi$ are both smaller than the those in the classic linear regret bound, because the denominator term $\lambda_{\min}(V_{T_0})$ is larger than the denominator term $\lambda$ in the classic linear regret bound. Overall, if $\Delta_\alpha(T-T_0) \leq B_T\sqrt{2(T-T_0)\,\Psi}$, or, equivalently, $T - T_0 \leq B_T^2/\Delta_\alpha^2\,\Psi$, then the post-exploration phase is leading to smaller errors. This says that, while the long-term rate of the regret of Algorithm 2 coincides with the classic linear bandit regret bound, the short-term rate is better than the classic linear bandit regret bound. Moreover, if the LLM is powerful in predicting the rewards with delicate fine-tuning and prompt engineering, we can benefit a lot from a small $\Delta_\alpha$. Third, the quality of the embedding-based prediction matters a lot, which is captured by the parameter $R$. When the embedding has high quality for encoding the semantic information of the task, then $R$ is small, and the embedding-based prediction is more accurate with a smaller $B_T$. Otherwise, the embedding-based prediction is less accurate with a larger $B_T$.

## 5 EXPERIMENTS

In this section, we aim to answer these questions:

- RQ1: Does the integrated LLM–MAB agent consistently outperform both classical bandit methods (e.g., LinUCBs) and pure LLM-based strategies (e.g., CoRRAL) across diverse recommendation benchmarks?

- RQ2: How do different settings of the embedding-based uncertainty weight $\beta_{\text{LLM}}$ and LLM sampling temperature affect the exploration–exploitation trade-off and final cumulative reward?

- RQ3: How many expensive LLM queries can be avoided by delegating decision rounds to a lightweight MAB component, while still preserving or improving cumulative reward?

## 5.1 Experimental Setups

### 5.1.1 Datasets.

The experiments are conducted using two publicly available datasets, LastFM (Cantador et al., 2011) and Movielens-1M (Harper & Konstan, 2015), and to evaluate the proposed method in diverse settings. Following Chen et al. (2024c), we choose the LastFM and Movielens-1M datasets. LastFM (Cantador et al., 2011): A dataset collected from a radio listening application, containing user play history and metadata of songs. This dataset is particularly useful for evaluating recommendation systems. Movielens-1M (Harper & Konstan, 2015): A dataset derived from the MovieLens project, which contains 1 million movie ratings from 6,040 users on 3,952 movies. It serves as a benchmark for collaborative filtering and recommendation algorithms. We only use them for research purposes.

### 5.1.2 Baselines

To provide a comprehensive comparison, several state-of-the-art baselines are selected:

- Single LinUCB (Li et al., 2010b): A linear upper confidence bound algorithm that provides a simple baseline for the bandit setting.
- LinUCB (Li et al., 2010b): The classic linear UCB algorithm with improved performance compared to the single version.
- Linear Bandit (AdamLinear) (Foster & Rakhlin, 2020): An adaptive gradient method applied in the context of linear bandits, known for its robustness and efficiency.
- LLM baseline (LLama 3.2 3B (Grattafiori et al., 2024)): A generative pre-trained model used as a basic comparison point for natural language processing tasks integrated into decision-making agents.
- Uncertainty-based LLM (Tanneru et al., 2024) (varying temperature hyperparameters): The same GPT baseline but with different temperature hyperparameters to explore the impact of uncertainty in decision-making.
- CoRRAL (Chen et al., 2024a): A recent algorithm that leverages adaptive mechanisms for efficient exploration and exploitation, serving as a strong comparative baseline.

## 6 Implementation Details

For conducting experiments, we use a single NVIDIA A100 GPU equipped with 80GB of GPU memory. For Movielens dataset, we set the total rounds $T = 6{,}500$, exploration phase $\sigma T = 1{,}000$, regularization $\lambda = 1.0$, and LLM-uncertainty weight $\beta_{\text{LLM}} = 0.8$ (selected via validation). All methods are run for 5 seeds, and results are averaged. For LastFM dataset, we set the total rounds $T = 2{,}500$, exploration phase $\sigma T = 1{,}000$, regularization $\lambda = 1.0$, and LLM-uncertainty weight $\beta_{\text{LLM}} = 0.8$ (selected via validation). All methods are run for 10 seeds, and results are averaged. For prompt-based reward extraction, we employ the `Llama 3.2-3B-Instruct` model to compute a reward score for each sample, where each sample consists of a context–item pair. The details of prompt design refer to Section 6.1. For embedding-based uncertainty quantification, we use logistic regression configured with L2 penalty, regularization strength $\lambda = 1.0$, the saga solver, and a maximum of 100 iterations. All input embeddings are first standardized to zero mean and unit variance via StandardScaler. For fair comparison, in all our experiments, we select LLM backbones from the `Llama 3.2-3B-Instruct` model (Grattafiori et al., 2024).

### 6.1 Prompt Design

For the Movielens dataset:

- **System message:** You are a movie recommender system specialized on the MovieLens dataset.
- **User message:**
  User watching history: {query}
  Candidate movies: {candidates}
  Recommend exactly one movie by outputting only its name (no explanations).

For the LastFM dataset:

- **System message:** You are a music recommender system specialized on the Last.FM dataset.
- **User message:**
  User listening history: {query}
  Candidate tracks: {candidates}
  Recommend exactly one track by outputting only its name (no explanations).

**Evaluation metrics.** To comprehensively evaluate the performance of our proposed model, we measure its effectiveness using the ***Average Cumulative Rewards***: The mean total reward accumulated per episode (or round) across all runs, reflecting both the quality of individual decisions and the long-term performance stability.

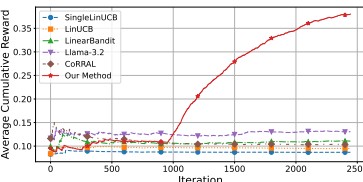

Figure 2: LastFM Cumulative Reward.

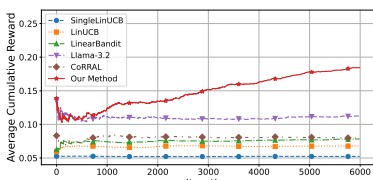

Figure 3: MovieLens Cumulative Reward.

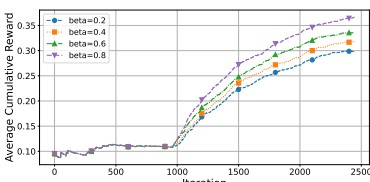

Figure 4: LastFM $\beta_{LLM}$ Ablation.

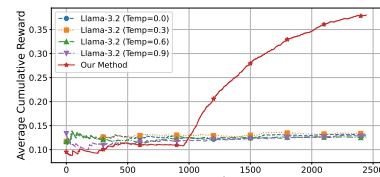

Figure 5: LastFM Uncertainty.

Table 1: Average Cumulative Reward on the ML-1M and LastFM benchmarks. The symbol ↑ indicates that higher values are better.

| Method | Average Cumulative Reward (↑) | |
| --- | --- | --- |
| | LastFM | ML-1M |
| SingleLinUCB | 0.0913 | 0.0528 |
| LinUCB | 0.0993 | 0.0683 |
| LinearBandit | 0.1269 | 0.0783 |
| Llama 3.2 | 0.1386 | 0.1350 |
| CoRRAL | 0.1511 | 0.0843 |
| *Our Method* | **0.3801** (+151.6%) | **0.1846** (+36.7%) |

As shown in Table 1, our method dramatically outperforms all baselines on the LastFM and Movielens-1M benchmarks. Also, Figure 2 and Figure 3 illustrate that *our method* consistently achieves the highest average cumulative reward after about 1000 rounds on both LastFM and Movielens-1M from the per-iteration cumulative reward curves, outperforming classical bandit baselines as well as LLM-based strategies. The first 1000 rounds can be viewed as a warm-up phase, during which the offline-trained model adapts to the online environment using observed feedback. Notably, our method not only accumulates rewards more rapidly but also maintains greater stability throughout the exploitation phase after this initial period. This is attributed to the early elimination of suboptimal actions and the effective integration of LLM-derived prompt rewards with embedding-based uncertainty estimates.

Together, these components enable informed exploration in the early stages and efficient exploitation thereafter, resulting in robust and consistent performance across both recommendation tasks.

## 6.2 UNCERTAINTY COMPARISON (RQ2)

All experiments in this section follow the online integration procedure from Algorithm 2. We focus on two aspects of uncertainty modeling: the weight of the embedding-based uncertainty term $\beta_{\text{LLM}}$, and the effect of LLM sampling temperature when used standalone as a decision agent compared to our method. Figure 3 plots the average cumulative reward on LastFM for different values of $\beta_{\text{LLM}}$. When $\beta_{\text{LLM}}$ is very small, the agent under-explores, relying almost entirely on the prompt-derived reward prior. As $\beta_{\text{LLM}}$ increases, the learned uncertainty estimator kicks in, guiding more balanced exploration and yielding substantial gains. Beyond the optimal regime, further increasing $\beta_{\text{LLM}}$ yields diminishing returns, as over-emphasis on uncertainty can lead to excessive exploration. This ablation confirms that a properly calibrated uncertainty weight is crucial for robust performance. Figure 4 compares *our method* against the LLaMA 3.2 3B baseline run with four different temperature settings ($\tau \in \{0, 0.3, 0.6, 0.9\}$). While temperature tuning slightly affects the baseline's exploration behavior, all LLaMA variants plateau early and fail to match the upward trajectory of *our method*. In contrast, by explicitly combining prompt rewards with an embedding-based uncertainty signal, our approach continues to drive accurate exploration throughout the decision rounds, achieving increasingly higher cumulative rewards afterwards. This demonstrates that static temperature adjustments alone cannot substitute for a principled uncertainty estimator in bandit decision-making.

## 6.3 COMPUTATIONAL EFFICIENCY (RQ3)

Table 2: LLM and MAB call statistics on MovieLens-1M and LastFM. $CP$ means call percentage.

| Statistic | CoRRAL | | Our Method | |
|---|---|---|---|---|
| | *Number* | *CP(%)* | *Number* | *CP(%)* |
| ML-1M LLM calls | 3000 | 49.67 | 1000 | 16.56 |
| LastFM LLM calls | 1500 | 60.12 | 1000 | 40.08 |

Table 2 demonstrates that, unlike CoRRAL, which splits decision rounds roughly equally between LLM and MAB, *our method* confines LLM calls to an initial exploration phase and delegates the vast majority of subsequent decisions to the lightweight MAB component. By reducing LLM usage by over 20% compared to CoRRAL, we cut API latency and cost substantially, while still achieving superior cumulative reward.

## 7 CONCLUSION

In this paper, we introduce a lightweight framework that fuses prompt-derived priors from an LLM with an embedding-based uncertainty estimator in a contextual bandit. On LastFM and ML-1M, our method achieves state-of-the-art cumulative reward, cuts LLM queries by over 20%, and maintains a strong exploration–exploitation balance.

## LIMITATIONS

Our approach assumes access to an open-source LLM that exposes internals such as embeddings. In cases where only closed-source or black-box APIs are available, obtaining these representations may not be straightforward. Adapting our method to work with limited API access, for example, using proxy embeddings or lightweight adapter modules, could introduce additional engineering overhead but is unlikely to affect the core algorithmic insights.

ETHICAL CONSIDERATIONS

By leveraging a large pretrained language model, our method inherits potential biases present in its training data, which could lead to unfair or harmful recommendations.

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

## A  CASE STUDY

In this case study, we compare three recommendation strategies for a 25-year-old male user whose watch history includes over twenty films such as Son in Law, Beetlejuice, Aliens, Escape from L.A., and Lord of the Flies. Presented with the same set of twenty candidate movies, the pure LLaMA 3.2 model mistakenly selects Bloody Child, and the classic LinUCB algorithm chooses Thinner, whereas our LLM-calibrated LinUCB approach correctly ranks Star Wars: Episode VI – Return of the Jedi at the top. This choice is accurate for two complementary reasons. First, Star Wars: Episode VI shares core sci-fi and adventure elements with several of the user's past favorites (Aliens and Escape from L.A.), so the LLM's semantic scoring naturally favors it. Second, during the online exploration phase, our method's UCB adjustment identifies that Star Wars consistently yields higher real-time reward feedback than other high-scoring candidates, confirming its true preference alignment. In contrast, a standalone LLM can only rely on static semantic similarity (and so can be misled by less relevant titles), and a standalone bandit method lacks that rich prior, leading to suboptimal initial choices. By combining LLM priors with principled exploration, our hybrid method both recognizes the user's genre tastes and rigorously validates them through interaction.

---

**Comparison at LLama 3.2, LinUCB, and our method**

**Query:**
This user's age is 25, gender is Male; History:
Son in Law | Beetlejuice | Aliens | Escape from L.A.|...| Cutting Edge, The | Young Poisoner's Handbook, The | Lord of the Flies

**Candidates:**
- Full Speed
- My Boyfriend's Back
- Shooting Fish
- Mr. Jealousy
- Gold Diggers: The Secret of Bear Mountain
- Star Wars: Episode VI – Return of the Jedi
- Little Odessa
- Twin Peaks: Fire Walk with Me
- Second Best
- Wisdom
- Dazed and Confused
- American Werewolf in London, An
- Ulee's Gold
- Children of a Lesser God
- D3: The Mighty Ducks
- The Siege
- Out of Sight
- Cutting Edge, The
- The Young Poisoner's Handbook
- Lord of the Flies

**Ground Truth:**
Star Wars: Episode VI – Return of the Jedi

**Choices:**

1) Llama 3.2 output: Bloody Child

2) LinUCB output: Thinner

3) Our Method output: Star Wars: Episode VI – Return of the Jedi

---

## B  THE USE OF LARGE LANGUAGE MODELS (LLMS)

In preparing this manuscript, we leveraged large language models (LLMs) as writing assistants to improve readability and clarity. Specifically, LLMs were employed to polish sentences and

adjust phrasing to align with the conventions of academic writing. Importantly, the scientific ideas, methodological details, and results presented in this paper are entirely original and were conceived, designed, and validated by the authors. The use of LLMs was limited to language refinement rather than content generation, ensuring that all technical contributions remain the authors' own.

## C  THEORETICAL ANALYSIS

**Lemma 1** (Uniform score envelope). *Fix $\alpha \in (0,1)$. Define*

$$\Delta_\alpha := b + \sigma_s \sqrt{2 \log\left(\frac{A_{\max} T}{\alpha}\right)}.$$

*Then, with probability at least $1 - \alpha$,*

$$\max_{1 \leq t \leq T} \max_{a \in \mathcal{A}_t} |s_t(a) - \mu_t(a)| \leq \Delta_\alpha.$$

*Proof of Lemma 1.* For any fixed $(t, a)$, conditional subgaussianity and Hoeffding-type tails give $\Pr\left(|\xi_t(a)| > \tau \mid \mathcal{F}_{t-1}\right) \leq 2e^{-\tau^2/(2\sigma_s^2)}$. Adding bias $b_t(a)$ with $|b_t(a)| \leq b$ yields $\Pr\left(|s_t(a) - \mu_t(a)| > b + \tau \mid \mathcal{F}_{t-1}\right) \leq 2e^{-\tau^2/(2\sigma_s^2)}$. Union bound over all $(t, a)$ and set $\tau = \sigma_s \sqrt{2 \log(A_{\max} T/\alpha)}$ to obtain the claim. ∎

Lemma 1 quantifies the accuracy of the LLM prediction. It depends on two aspects: the inherent bias $b$ and the randomness of the LLM prediction $\sigma_s$. $\sigma_s$ can usually be reduced by using lower temperature. $b$ depends on many factors, such as prompt, model, etc.

*Proof of Theorem 1.* **Exploration Phase.** Conditioned on the event $\mathcal{E}_\alpha$ (Lemma 1), at any $t \leq T_0$, since $a_t = \arg\max s_t(\cdot)$ and $|s_t - \mu_t| \leq \Delta_\alpha$,

$$\mu_t(a_t^\star) - \mu_t(a_t) \leq s_t(a_t^\star) + \Delta_\alpha - \left(s_t(a_t) - \Delta_\alpha\right) \leq 2\Delta_\alpha.$$

Summing yields the term $2\Delta_\alpha T_0$.

**Post-exploration Phase.** We will use a similar argument as Abbasi-Yadkori et al. (2011b). Let

$$\beta_t = R\sqrt{2 \log \frac{\det(V_{t-1})^{1/2}}{\det(V_{T_0})^{1/2}} \frac{1}{\delta_0}} + \sqrt{\lambda} S,$$

and

$$w_t(a) = \|e_t(a)\|_{V_{t-1}^{-1}}.$$

Abbasi-Yadkori et al. (2011b) proved that

$$\beta_t \leq B_T = R\sqrt{2\left(d \log(1 + \frac{(T-T_0)L^2}{d \lambda_{\min}(V_{T_0})}) + \log \frac{1}{\delta_0}\right)} + \sqrt{\lambda} S.$$

By the self-normalized inequality with prior $V_{T_0}$ (Abbasi-Yadkori et al., 2011b), with probability $\geq 1 - \delta_0$, for all $t > T_0$ and all $a$, $|e_t(a)^\top (\widehat{\theta}_{t-1} - \theta^\star)| \leq \beta_t w_t(a)$, hence $\mu_t(a) \in [\text{LCB}_t(a), \text{UCB}_t(a)]$. By the selection rule, if $a_t$ is selected, $r_t(a_t) \geq r_t(a_t^\star)$.

On one hand, if $\Delta_\alpha \geq \beta_t w_t(a_t)$, we have

$$\mu_t(a_t^\star) - \mu_t(a_t) \leq r_t(a_t^\star) - r_t(a_t) + 4\beta_t w_t(a_t) \leq 4\beta_t w_t(a_t) \qquad \text{for all } t > T_0. \tag{7}$$

On the other hand, if the LLM prediction accuracy $\Delta_\alpha$ is smaller than the bound $\beta_t w_t(a_t)$, we can prove that

$$|r_t(a_t) - \mu_t(a_t)| = |r_t(a_t) - s_t(a_t) + s_t(a_t) - \mu_t(a_t)| \tag{8}$$

If $r_t(a_t) = s_t(a_t)$, then $|r_t(a_t) - \mu_t(a_t)| \leq \Delta_\alpha$ by Lemma 1. If $r_t(a_t) < s_t(a_t)$, then $0 < s_t(a_t) - r_t(a_t) \leq \mu_t(a_t) + \Delta_\alpha - r_t(a_t) \leq \Delta_\alpha$, hence $|r_t(a_t) - \mu_t(a_t)| \leq \Delta_\alpha$. Similarly, when $r_t(a_t) > s_t(a_t)$, we can prove the same bound. Therefore, in this case, we always have $|r_t(a_t) - \mu_t(a_t)| \leq \Delta_\alpha$. By the selection rule of Algorithm 2, we have

$$r_t(a_t) \geq \text{clip}(s_t(a_t^\star), \text{LCB}_t(a_t^\star), \text{UCB}_t(a_t^\star)).$$

- If $\text{clip}(s_t(a_t^\star), \text{LCB}_t(a_t^\star), \text{UCB}_t(a_t^\star)) = s_t(a_t^\star)$, then $r_t(a_t) \geq s_t(a_t^\star)$.

- If $\text{clip}(s_t(a_t^\star), \text{LCB}_t(a_t^\star), \text{UCB}_t(a_t^\star)) = \text{LCB}_t(a_t^\star)$, then $r_t(a_t) \geq \text{LCB}_t(a_t^\star) \geq s_t(a_t^\star)$.

- If $\text{clip}(s_t(a_t^\star), \text{LCB}_t(a_t^\star), \text{UCB}_t(a_t^\star)) = \text{UCB}_t(a_t^\star)$, then $r_t(a_t) \geq \text{UCB}_t(a_t^\star) \geq \mu_t(a_t^\star) \geq s_t(a_t^\star) - \Delta_\alpha$.

To conclude, if $a_t$ is selected and $\Delta_\alpha$ is smaller than the bound $\beta_t w_t(a_t)$, then we have

$$s_t(a_t^\star) - r_t(a_t) \leq \Delta_\alpha, \quad \mu_t(a_t^\star) - \mu_t(a_t) \leq 3\Delta_\alpha.$$

Therefore, the per-round regret is bounded by

$$\mu_t(a_t^\star) - \mu_t(a_t) \leq 4 \min(\Delta_\alpha, \beta_t w_t(a_t)).$$

The above analysis gives

$$\sum_{t=T_0+1}^{T} \left(\mu_t(a_t^\star) - \mu_t(a_t)\right) \leq \sum_{t=T_0+1}^{T} 4 \min(\Delta_\alpha, \beta_t w_t(a_t))$$

$$\leq \min\left\{ \sum_{t=T_0+1}^{T} 4\Delta_\alpha, \sum_{t=T_0+1}^{T} 4\beta_t w_t(a_t) \right\}.$$

The first summation in the minimum is simply $4\Delta_\alpha(T - T_0)$. The second summation in the minimum is bounded by $4B_T \sqrt{(T - T_0)\,\Psi}$, following the classic argument in Abbasi-Yadkori et al. (2011b). Therefore, the per-round regret is bounded by

$$\sum_{t=T_0+1}^{T} \left(\mu_t(a_t^\star) - \mu_t(a_t)\right) \leq \min\left\{ \sum_{t=T_0+1}^{T} 4\Delta_\alpha, \sum_{t=T_0+1}^{T} 4\beta_t w_t(a_t) \right\}$$

$$\leq 4 \min\left\{ \Delta_\alpha(T - T_0)\,,\ B_T\sqrt{2(T - T_0)\,\Psi} \right\}.$$

Taking the minimum of the two post-exploration controls completes equation 6.

∎

