# OpenReview forum: "Contextual Bandits with LLM-Derived Priors and Adaptive Calibration"
_ICLR.cc/2026/Conference — ICLR 2026 Conference Withdrawn Submission_

### Official Review · Reviewer_q8gv · 2025-10-20

**Soundness:** 1
**Presentation:** 1
**Contribution:** 2
**Rating:** 2
**Confidence:** 3

**Summary:**

This paper proposes a contextual bandit framework that integrates LLM-derived semantic priors with a standard linear bandit model. The main idea is to use the LLM’s output to estimate a prior reward for each arm, then refine and constrain it with uncertainty estimates and confidence bounds from both an embedding-based reward predictor and a linear bandit estimator. The method aims to combine semantic knowledge from language models with online learning to improve exploration–exploitation trade-offs. Experiments on recommendation-style datasets show performance improvements over standard bandit baselines and LLM baseline.

**Strengths:**

1. The core idea of combining LLM-derived semantic priors with a contextual bandit framework is novel and interesting, and it addresses an important problem of integrating language understanding with online decision-making.
2. The paper provides theoretical analysis and regret guarantees, which add credibility to the proposed method and strengthen its technical contribution.

**Weaknesses:**

1. Extremely poor writing and presentation quality. The paper is very hard to follow due to numerous writing issues and careless mistakes, which give the impression of a sloppy and even irresponsible attitude. For example, Figure 1 is incomplete (the word “reward” is cut off to “rewa”), and in Eq. (4) the variable v is written as u, which is confusing and unprofessional.
2. Algorithm 2 is severely underspecified and inconsistent. In the step “Compute LLM reward prediction,” all three rewards are denoted as r_LLM, making it impossible to distinguish their meaning. Eq. (5) claims that the clipping step uses the confidence interval computed from the last-layer feature e(x), but Algorithm 2 instead uses the confidence interval based on x — a direct inconsistency. The paper never explains how θ is updated. It also fails to explain why a second set of UCB/LCB bounds appears at all, nor how they relate to the embedding-based ones. Moreover, the authors never specify how the contextual vector x is computed from the raw text, which is a critical detail for reproducibility.
3. Experimental analysis is incomplete. From the results, it is clear that the exploration-phase term σT is crucial for the performance of the proposed method. However, the paper does not include any ablation study on the σ parameter, which is necessary to understand its impact and validate the claimed benefits.
4. Choice of LLM severely limits the practical value of the work. The experiments are conducted only with a small LLM (Llama 3.2 3B), which raises concerns about the scalability and applicability of the proposed method. No ablation study is provided to demonstrate whether the approach still works when larger models are used.

**Questions:**

1. Prompt details: Please provide the prompt that queries the LLM for an evaluation of the expected reward. This information is essential for reproducibility.
2. σ ablation: Since the exploration-phase variance term σ clearly affects performance, can you provide an ablation study showing how different σ values influence results?
3. Algorithm 2 involves two sets of UCB/LCB bounds, but their roles are unclear. Please clarify how each is updated and how they are used during action selection.
4. Please report results for two ablations: (a) selection based only on r_prompt; and (b) selection based only on r_est. This will help quantify the contribution of each component.

---

### Official Review · Reviewer_Q9zQ · 2025-10-27

**Soundness:** 2
**Presentation:** 2
**Contribution:** 2
**Rating:** 2
**Confidence:** 4

**Summary:**

This paper proposes a hybrid framework that integrates LLMs with contextual bandits. The idea is to use LLMs as "priors" for reward prediction and uncertainty estimation, combining both:

1. **Prompt-derived rewards.** Zero-shot supervision from the LLM’s responses,
2. **Embedding-based uncertainty.** Computing the (standard, LinUCB-style) elliptical (self-normalized) confidence set based on the LLM’s emebddings.

An online adapter recalibrates estimates and their uncertainty to handle distribution shifts. The method is evaluated on the MovieLens-1M and LastFM datasets, showing higher cumulative rewards (basically a jump in rewards once the first phase is done) and fewer LLM calls. A regret bound is derived that interpolates between the biased LLM regime and the standard linear-bandit setting.

**Strengths:**

**S1.** The paper tackles an important topic: integrating pretrained LLMs with online decision-making.

**S2.** Presents a grounded framework combining LLMs with classical bandit confidence bounds.

**S3.** Includes a formal regret bound and reasonable empirical results.

**Weaknesses:**

**W1. Unclear presentation and notation.**
The overall flow is reasonable, but frequent notation inconsistencies and missing explanations make deep understanding and reproducibility difficult.
For example, the paper uses $b_t$ both as (i) the linear-bandit regressand vector in Algorithm 2 and (ii) a scalar LLM bias $b_t(a)$ in Assumption 2.
Although one is boldfaced and the other is not, reusing the same symbol for two unrelated quantities makes the paper unnecessarily hard to follow.
Likewise, $e_t(x_t)$, $e_t(x_t, r_t)$, $e(a)$, and other variants appear throughout, though only one form $e(x, c)$ is clearly and mathematically defined.
Contexts $x_{t,i}$ in Algorithm 2 are treated both as text (inputs used to extract embeddings) and as vectors (to update $G_t$ and $b_t$).
$r_{\text{LLM}}$ is assigned three different meanings within Algorithm 2 without clarifying which value is used at each step. Also, what is the definition of $\hat \theta_t$, $\hat \theta_t = G_t^{-1} b_t$?
Overall, a thorough pass is needed to unify notation, explicitly define every variable at first use, and ensure consistency between the algorithmic and theoretical parts.

---

**W2. Unclear computational efficiency and LLM-usage assumptions.**
The benefits of reducing “LLM calling” is not clear to me.
After the warm-up phase, text generation stops, but embeddings are still extracted each round.
When LLM outputs are only a few words (e.g., movie title), it is not evident why generation would be much slower than embedding extraction, especially when using KV caching.
The approach would indeed be cheaper online if embeddings were precomputed offline, but this requires the set of contexts to be fixed in advance: a strong assumption not discussed.
If embeddings are instead obtained online via LLM forward passes, the savings over pure-LLM calling might not be huge.
A fair comparison should report wall-clock runtimes of pure-LLM calling vs embeddings caling.

---

**W3. Incremental theoretical novelty.**
The main theorem is a direct adaptation of the elliptical-confidence-set lemma with an additive LLM bias term.
While formally correct, it offers limited conceptual novelty (regarding the usage of LLMs) beyond standard linear-bandit results.
Collapsing the entire LLM contribution into a single additive bias term feels overly simplistic.

---

**W4. Need for more comprehensive ablations.**
Given the modest theoretical contribution, the empirical evaluation should more clearly disentangle the effect of each module. No ablations are provided that remove or vary the prompt reward $r_{\text{prompt}}$, the embedding-based uncertainty $\nu(x)$, or the online contextual adapter: each of which is an important component of the proposed method. Even for the only ablated parameter, $\beta_{\text{LLM}}$, the reported range stops at 0.8, despite claims of diminishing returns beyond that value.

---


**W5. Missing or limited baselines.** The set of baselines remains relatively narrow, omitting several comparisons:

* Heuristic LLM uncertainty estimates (e.g., response entropy, sampling diversity, etc.).

* Standard bandit algorithms (LinUCB, Thompson Sampling) using LLM embeddings directly as features. In particular, it is unclear which feature representation LinUCB actually uses in the experiments.
* Thompson Sampling with pre-trained or informative priors, which directly aligns with the paper’s “LLM prior” motivation. While I am not aware of a prior work using LLM-based priors for TS, recent studies have explored diffusion-model priors for TS [1, 2], and discussing this line of work would be relevant. For experiments, a straightforward and meaningful baseline would be TS with Gaussian informative priors, where the prior mean is set to the LLM embedding of each movie (or item) and the prior variance is tuned to balance exploration. Such a simple baseline would likely perform competitively on MovieLens.

[1] https://arxiv.org/pdf/2402.10028 (Neurips 2025)

[2] https://arxiv.org/pdf/2410.03919 (Neurips 2024)

---

**W6. Incomplete pipeline description and limited reproducibility.**
The implementation section omits details that are necessary to reproduce the results. Baseline hyperparameters are unspecified (e.g., does LinUCB use a tuned $\beta_t$?), and the procedure for computing prompt embeddings, whether averaging token embeddings or using the final token (EOS) embedding, is not described. Moreover, the feature representation used by LinUCB is unclear.

**Questions:**

**Q1. Clarification on embeddings and context usage.**

   * Are embeddings extracted online at every round or precomputed offline?
   * If they are precomputed, how is this compatible with dynamic or unseen contexts?
   * In Algorithm 2, are contexts \$x_{t,i}\$ treated as text inputs (to get embeddings) or as vectors (to update $G_t$ and $b_t$)?

**Q2. Definition and role of $r_{\text{LLM}}$.**

   * Algorithm 2 assigns three different quantities to $r_{\text{LLM}}$. How each of these quantities is exactly used?

**Q3. Online adapter mechanism.**

   * Does the adapter update both the prediction head $\varphi$ and the covariance matrix $V_t$, or only the uncertainty term?

**Q4. Confidence-width calibration.**

   * $\beta_{\text{LLM}}$  was tuned, are the hyperparameters of LinUCB and CoRRAL baselines also tuned?


**Q5. Computation and efficiency claims.**

   * Have you measured wall-clock runtime (not just the number of LLM calls) for your approach compared to CoRRAL and pure-LLM baselines?
   * When answers are short, which is the case here (e.g., answer is song/movie name), how much faster is embedding extraction in practice relative to text generation?

**Q6. Ablations and component importance.**

   * Could you provide results that remove or vary key components individually: e.g., the prompt reward $r_{\text{prompt}}$, the embedding-based uncertainty $\nu(x)$, and the online adapter?
   * How does performance change when each of these is disabled?

**Q7. Prompt-embedding computation.**

   * How are prompt embeddings computed exactly? LLMs output an embedding per token and hence how you use that to compute the overall prompt embedding:  average over token embeddings, final-token embedding?

---

### Official Review · Reviewer_g2ZQ · 2025-10-27

**Soundness:** 2
**Presentation:** 1
**Contribution:** 2
**Rating:** 2
**Confidence:** 4

**Summary:**

The paper investigates how to integrate prior knowledge from LLMs into contextual bandit problems, specifically to address the lack of principled exploration mechanisms. The authors propose a lightweight framework that integrates LLM-derived priors with an adaptive calibration mechanism within a multi-armed bandit (MAB) setting. The methodology first involves extracting prompt-based rewards from the LLM to provide task-specific supervision. Concurrently, it constructs an uncertainty estimator based on the LLM's penultimate-layer embeddings, which uses a linear projection head and an empirical covariance matrix to quantify uncertainty and create confidence bounds. To handle distribution shifts, this framework also incorporates an online contextual uncertainty adapter to dynamically update these estimates during interaction. The online integration (Algorithm 2) follows a two-phase approach: an initial exploration phase (for $t \le \sigma T$) relies solely on the LLM's reward estimate, after which a joint phase selects actions using the LLM's prompt reward clipped within the confidence bounds (LCB and UCB) calculated by a standard MAB component. The paper evaluates this method on the LastFM and MovieLens-1M datasets, using Average Cumulative Reward as the primary metric and comparing against baselines such as LinUCB and CORRAL. Based on these experiments, the paper reports that the proposed method achieved higher cumulative rewards than the baselines on both datasets , while also substantially reducing the number of expensive LLM queries. The authors also provide theoretical regret guarantees, which they state establish improved sample efficiency over standard contextual bandit algorithms.

**Strengths:**

* The work introduces an innovative framework that principledly integrates LLM's contextual priors with classical bandit algorithms, the Upper/Lower Confidence Bound to adaptively calibrate and clip the LLM's contextual predictions.
* The work provides a the regret bound analysis for the proposed algorithm.
* The work conducts experiments on two prevalent recommendation system datasets to validate the algorithm's performance through a comprehensive comparison with baseline methods.

**Weaknesses:**

This work suffers from significant issues in clarity, notation, and the overall description of the core algorithm, which severely hinders its readability and understanding. Several points require immediate clarification:

1.  **Confusing Algorithmic Logic and Notation:**
    * The paper's general writing style often lacks logical flow, making it difficult to follow the proposed methodology.
    * The description of reward prediction in **Algorithm 2, Line 6, is particularly confusing**, as it presents three different $r_{LLM}(i)$ predictions without specifying which is used: (1) a purely prompt-based reward `Prompt` (potentially in-context learning, which lacks necessary details on prompt design in the paper); (2) an embedding-based UCB prediction; and (3) an embedding-based LCB prediction. The subsequent clipping step, which uses an *online MAB* derived UCB/LCB, adds another layer of confusion, leaving the reader to guess which of the three LLM scores is the target for calibration.

2.  **Ambiguity of Feature Vectors in MAB Component:**
    * In **Algorithm 2, Lines 7 and 8**, which compute the UCB and LCB for the MAB component, the input feature vector **$x_{t,i}$** is used. While the notation indicates it is a $d$-dimensional vector, the paper fails to clearly define what content this vector represents.

3.  **Contradiction Between Formal Equation and Algorithm Implementation:**
    * There appears to be a notable contradiction or inconsistency between the **formal reward definition in Equation (5)** and its implementation description in **Algorithm 2, Line 12**.
        * Equation (5) suggests the reward is clipped using **LLM-derived uncertainty terms** (i.e., $r_{est}(x_i) \pm \nu(x_i)$ from Equation (4), calculated using the embedding $e(x)$ and matrix $V$).
        * However, Algorithm 2, Line 12, explicitly uses $CLIP_{[\mathbf{r_{LCB}(i)}, \mathbf{r_{UCB}(i)}]}$ where $r_{LCB}(i)$ and $r_{UCB}(i)$ are computed using the **separate MAB statistics $G_t$ and $b_t$** with the unexplained $d$-dimensional feature $x_{t,i}$ (from Lines 7 and 8).
    * This discrepancy, coupled with the previously mentioned clarity issues, makes it extremely difficult to reconcile the proposed theory with the implementation details, forcing the reader to constantly guess the authors' intent and significantly increasing the reading burden.

**Questions:**

1. To robustly validate the claimed superiority, particularly for systems utilizing rich, non-linear representations, the authors may include a comprehensive performance comparison against a strong non-linear contextual bandit baseline, such as NeuralUCB.

2. Could the authors demonstrate the effect of skipping the initial exploration phase entirely by setting $T_0 = 0$ (or $\sigma=0$)?

3. The paper's current presentation is severely hampered by **inconsistent notation and numerous ambiguous implementation details**. **The authors must dedicate a section in their rebuttal to comprehensively and definitively clarifying all these sources of confusion** to make the algorithm readable and the work reproducible.

---

### Official Review · Reviewer_V4z6 · 2025-10-31

**Soundness:** 2
**Presentation:** 1
**Contribution:** 3
**Rating:** 2
**Confidence:** 4

**Summary:**

The paper “Contextual Bandits with LLM-Derived Priors and Adaptive Calibration” addresses the lack of principled exploration mechanisms when applying large language models (LLMs) to contextual bandits. The authors propose a lightweight framework that integrates LLM-derived priors with adaptive calibration to improve exploration and decision-making efficiency. The method extracts prompt-based rewards from LLMs as task-specific supervision, estimates uncertainty through embedding-based representations, and introduces an online contextual adapter for updates under distribution shifts. The paper provides theoretical regret bounds and validates the approach on LastFM and MovieLens datasets.

**Strengths:**

Positive Points
1.Novel and timely topic.
The paper explores integrating the semantic priors of large language models (LLMs) into the contextual bandit learning framework, which represents an important and emerging direction at the intersection of LLMs × online learning. The topic is timely and potentially impactful, especially for applications such as recommendation systems and interactive decision-making.
2.Conceptually interesting approach.
The authors propose using LLM-generated contextual scores as prior information and introduce an adaptive calibration mechanism through uncertainty estimation and reward clipping within the bandit framework. Although the implementation remains preliminary, the idea of combining language-informed priors with statistical exploration strategies is insightful and has the potential to inspire future research.

**Weaknesses:**

Weaknesses and Suggestions
Motivation
The work introduces an “online contextual adapter” to handle distributional shift, yet no experiments are conducted under non-stationary conditions. Although the motivation repeatedly emphasizes robustness to shift, all benchmarks used are static, leading to a clear mismatch between motivation and evidence.
Related Work
The related work section lacks coverage of recent research. The work by De Curtò J, de Zarza I, Roig G, et al., titled "LLM-informed multi-armed bandit strategies for non-stationary environments" (Electronics, 2023, 12(13): 2814), should be included in the references. This paper makes significant contributions to multi-armed bandit strategies in non-stationary environments. Additionally, the work by Sun J, Wang Z, Yang R, Xiao C, Lui J C S, and Dai Z, titled "Large Language Model-Enhanced Multi-Armed Bandits" (2025, arXiv: 2502.01118), should also be cited. This research explores the integration of large language models with multi-armed bandit algorithms, offering further insights into the enhancement of bandit strategies.
Assumptions
Assumption 2 postulates that, conditioned on history, the LLM-generated scores follow a biased sub-Gaussian distribution. This is unrealistic in practice—prompt variations, candidate-set changes, temperature sampling, and multi-turn accumulation all violate this condition.
Furthermore, the relationship between the bias bound b and the temperature parameter is mentioned only qualitatively, without empirical estimation or theoretical proof. This significantly weakens the credibility of the theoretical analysis.
Methodology
1.Unclear reward normalization and calibration.
The paper does not specify how the LLM-generated “prompt-derived rewards” are normalized to the [0, 1] interval, whether normalization is performed per candidate set, or whether calibration methods such as Platt scaling, temperature scaling, or quantile mapping are applied. These details directly affect the scale consistency of the clipped rewards ( \tilde{r} ) when integrated with the UCB term, thereby impacting interpretability and reproducibility.
2.Inconsistent notation and formula errors.
In Algorithm 2, the same symbol rLLM(i) is reused to denote multiple different quantities (prompt output, confidence bounds, and final clipped reward), which severely harms readability and reproducibility. The notation, derivations, and formula numbering across the algorithm section need to be systematically revised.
Experiments
1. The experiments compare only with “Single LinUCB / LinUCB / AdamLinear / CoRRAL / LLM (temperature)” baselines, omitting canonical non-stationary methods (e.g., SW-LinUCB, Discounted-LinUCB, Change-point UCB) and modern deep variants such as NeuralUCB and NeuralTS.
2. The paper reports only “Average Cumulative Reward,” while figures show per-iteration cumulative curves and tables provide a single scalar value. It is unclear whether these represent average per-round rewards or cumulative averages. Moreover, key metrics such as regret, CTR@K, NDCG, or hit-rate, as well as significance tests and confidence intervals, are missing.
3. Figures 2, 3, and 5 only show the upward trend of the return curve for the method proposed in this paper, but do the other methods not exhibit growth?
Other Issues
The same reference (Li et al., 2010) appears twice as “2010a/2010b” with almost identical content. Some entries have inconsistent years or citation formats.
The content of this work is somewhat simplistic, and certain parts are rather trivial.

**Questions:**

See weaknesses.

---

### Note · Authors · 2025-11-23

I have read and agree with the venue's withdrawal policy on behalf of myself and my co-authors.